# Controlling thermal reactivity with different colors of light

Hannes A. Houck [1,2,3,4], Filip E. Du Prez [4] & Christopher Barner-Kowollik [1,2,3]

The ability to switch between thermally and photochemically activated reaction channels with an external stimulus constitutes a key frontier within the realm of chemical reaction control. Here, we demonstrate that the reactivity of triazolinediones, powerful coupling agents in biomedical and polymer research, can be effectively modulated by an external photonic field. Specifically, we show that their visible light-induced photopolymerization leads to a quantitative photodeactivation, thereby providing a well-defined off-switch of their thermal reactivity. Based on this photodeactivation, we pioneer a reaction manifold using light as a gate to switch between a UV-induced Diels–Alder reaction with photocaged dienes and a thermal addition reaction with alkenes. Critically, the modulation of the reactivity by light is reversible and the individually addressable reaction pathways can be repeatedly accessed. Our approach thus enables a step change in photochemically controlled reactivity, not only in small molecule ligations, yet importantly in controlled surface and photoresist design.

[1] School of Chemistry, Physics and Mechanical Engineering, Queensland University of Technology (QUT), 2 George Street, Brisbane QLD 4000, Australia. [2] Macromolecular Architectures, Institut für Technische Chemie und Polymerchemie, Karlsruhe Institute of Technology (KIT), Engesserstraße 18, 76131 Karlsruhe, Germany. [3] Institut für Biologische Grenzflächen, Karlsruhe Institute of Technology (KIT), Hermann-von-Helmholtz-Platz 1, 76344 Eggenstein-Leopoldshafen, Germany. [4] Polymer Chemistry Research Group, Centre of Macromolecular Chemistry (CMaC), Department of Organic and Macromolecular Chemistry, Ghent University, Krijgslaan 281 S4-bis, 9000 Gent, Belgium. Correspondence and requests for materials should be addressed to F.E.D.P. (email: filip.duprez@ugent.be) or to C.B.-K. (email: christopher.barnerkowollik@qut.edu.au)

To date, several spring-loaded and chemically orthogonal reactions—some of them meeting the stringent click chemistry requirements[1, 2]—had a profound impact on fields ranging from drug discovery and bio-conjugation[3, 4] to soft matter materials sciences[5, 6]. The defined control of chemical reactivity by external stimuli is critical for the design of adaptive, self-reporting, and programmable materials[7]. Specifically, temperature and light enable the on demand construction of (complex) macromolecules via so-called transclick[8, 9] and photoclick[10, 11] approaches.

Whereas both thermally and photochemically triggered reactions enable a selective on-switch over chemical transformations, light-induced methods are more defined and allow for high levels of spatial and temporal resolution. As a result, several photoinduced reactions—some of them carrying click characteristics—have been employed in precision site-selective processes, most prominently surface functionalization[12, 13], scaffold design via 3D laser lithography[14–17], and in vitro protein imaging[11, 18]. Critically, light-induced processes offer the potential to conduct sequential photochemical reactions in an orthogonal manner (so-called λ-orthogonality) by activating carefully selected chromophores, each at a distinct and highly specific wavelength[19, 20]. However, well-defined electronic transitions within the chromophore system upon radiation are required for λ-orthogonality. Initially, elegant combinations of variable photocleavage reactions—mainly based on o-nitrobenzyl[21–23], 3′,5′-dimethoxy-benzoin[21, 22], and/or coumarin derivatives[22, 23]—were utilized for wavelength-dependent uncaging of bioactive molecules[24, 25]. More recently, light-induced pericyclic reactions have been introduced into the realm of λ-orthogonality by us, enabling the site-selective synthesis of block copolymers[19, 26]. Although pathway independent λ-orthogonal systems are scarce, the emerging field of catalyst-free visible light-induced ligation protocols is advancing rapidly[27–30].

Interestingly, the exploitation of different colors of light as a gate to switch between thermally and photochemically activated reaction channels is almost non-explored and constitutes a key next frontier within the realm of reaction control[31]. Indeed, simply selecting the outcome of chemical one-pot processes by the absence or presence of a photonic field or by different wavelengths will allow for the development of, e.g., sub-diffraction photoresists[32–34] or multi-area-selective surface lithography. To date, only a few photoswitchable systems have been designed in which an external light stimulus can induce isomerization and thereby modulate the chromophore's reactivity, yet do not allow for their thermal reactivity to be switched off completely[35, 36]. Another elegant example of outer field control is the light-activated reaction manifold exploiting o-methylbenzaldehydes, which can either undergo a light-induced transformation to an o-quinodimethane (a photocaged diene), susceptible toward a Diels–Alder reaction with maleimides, or a thermal imine formation via a condensation reaction with amines[31]. However, while the reported manifold can switch between both reaction pathways, its thermal reactivity, again, cannot be fundamentally halted. It is thus critical in this context that simple and straightforward concepts are developed in which the thermal reactivity of molecules can completely be switched off by light.

Herein, we pioneer such a concept based upon the reversible photopolymerization of 1,2,4-triazoline-3,5-diones (TADs)—highly powerful click-substrates[37]—upon visible light irradiation and its subsequent UV light-induced reaction. We demonstrate in the context of trapping experiments that this polymerization leads to the quantitative photodeactivation of TADs, thereby providing a selective on- and off-switch over their thermal reactivity. This visible light-triggered deactivation is subsequently combined with the UV-induced TAD photoenol ligation chemistry of o-methylbenzophenones to provide an advanced λ-orthogonal reaction system. Finally, the careful selection of a thermal TAD-based reaction allows to design a reaction manifold in which light is used as a gate to selectively switch between a thermally and photochemically activated reaction channel, providing light-induced selectivity of the reaction outcome.

## Results

**Visible light photodeactivation of TAD.** While the proposed light-induced reaction manifold is theoretically simple (refer to Fig. 1), the ability to control the unique reactivity of

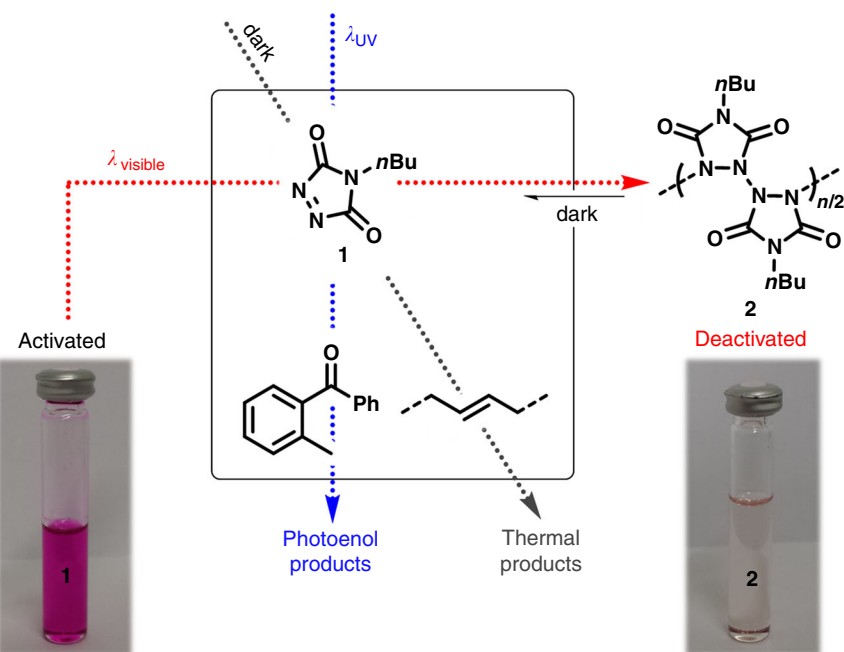

**Fig. 1** Schematic representation of the introduced light-controlled manifold. The photodeactivation of triazolinediones (TADs) under visible light irradiation enables a UV light-switchable reaction selectivity between a photoenol and thermal TAD-based reaction

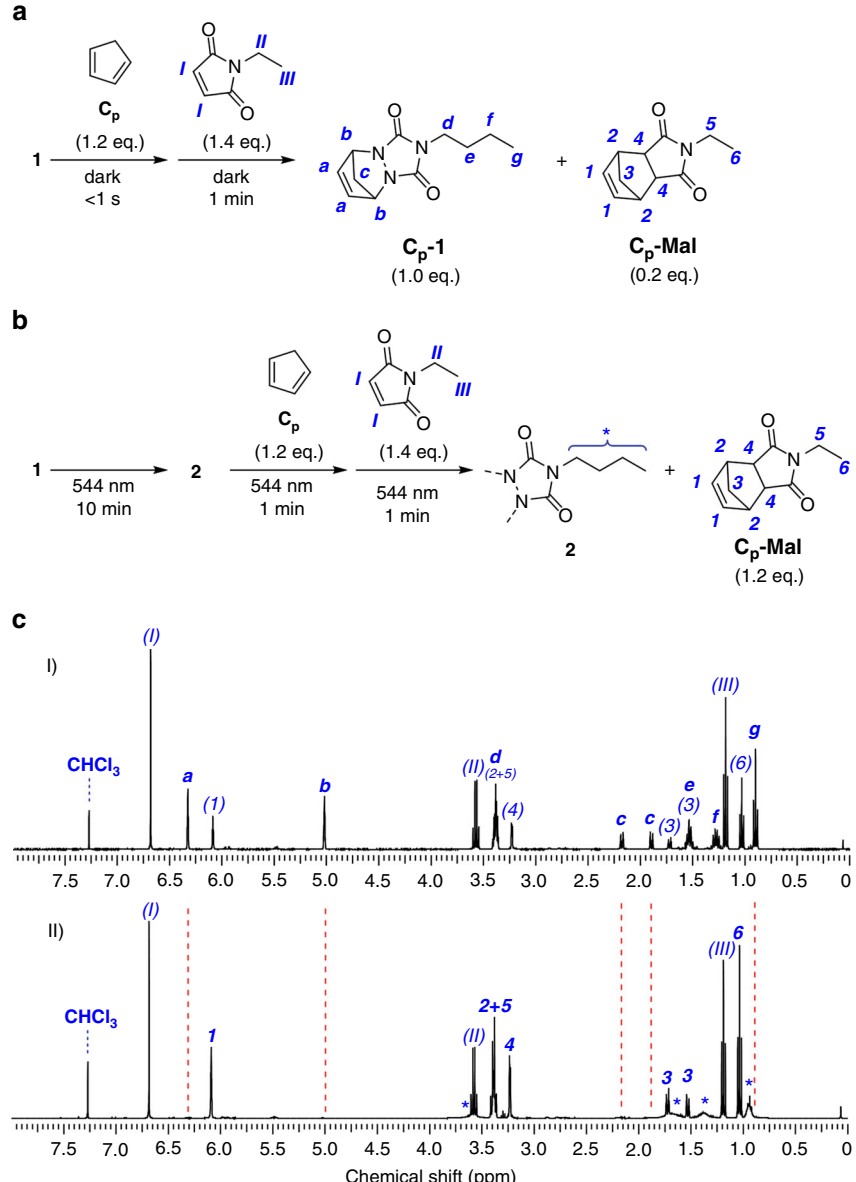

**Fig. 2** Proof of photodeactivation. **a** Trapping experiment consisting of the sequential addition of cyclopentadiene (1.2 eq.) and *N*-ethylmaleimide (1.4 eq.) to a solution of **1** (0.3 M in CDCl$_3$) either kept in the dark or, **b** continuously irradiated with visible light (544 nm, 4 mW cm$^{-2}$). **c** Whereas the $^1$H-NMR spectrum of the dark reaction (spectrum I) shows the complete trapping of **1** with cyclopentadiene, no such TAD adduct is observed when the visible light is switched on (II, dashed red line), thereby demonstrating the ability to photodeactivate **1** and thus to switch off the thermal TAD-based Diels–Alder reaction channel

triazolinediones is challenging since TAD-based ligation reactions often proceed very rapidly and are highly exergonic, even far below ambient temperature (e.g., −78 °C for reactions with cyclopentadiene)[38, 39]. Visible light irradiation even enhances the reactivity of TADs, making them eligible to undergo [4+2]-cycloaddition reactions with fullerenes[40], naphtalenes[41, 42], and even simple benzene derivatives[43, 44]. The starting point of our light-switchable reaction manifold was to exploit the observations by Pirkle and Stickler in the 1970s, who reported the exclusive homopolymerization of 4-*n*-butyl-TAD (**1**, cf. Fig. 1) under visible light irradiation to provide the all-nitrogen backbone polymer **2**[45]. Interestingly, **2** was found to slowly regenerate the initial red colored monomer over time, with up to 80% release of **1** after several days. While one avenue to photodeactivate a molecule is by changing its electronic properties thereby preventing a thermal chemical reaction to proceed, a viable

alternative is offered via a reversible light-induced transformation of the target substrate into a non-reactive moiety. We herein opt for the latter strategy and demonstrate that the photopolymerization of **1** can result in the photochemical deactivation of TADs.

Our initial experiments aimed at reproducing the observed formation of polymeric **2** under visible light irradiation. Since little details concerning the applied emission source were reported[45], we initially focused on establishing the wavelength at which the TAD polymerization can be effectively executed by making use of a wavelength-tunable laser system (Supplementary Fig. 1). Irradiation (4 mW cm$^{-2}$) of a 0.3 M deaerated purple solution of sublimed **1** in CCl$_4$ at the wavelength of maximum absorption, i.e., $\lambda_{max}$ = 544 nm (refer to the SI for the emission spectrum) gave a colorless solution after 10 min, which slowly regained its characteristic purple color upon standing in the dark

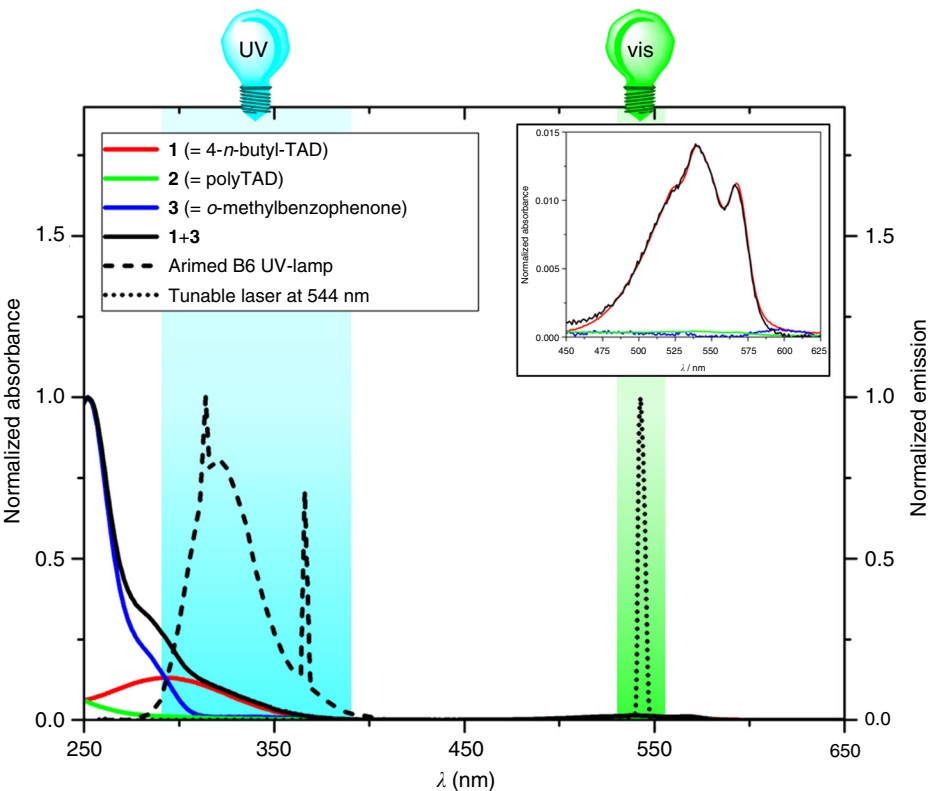

**Fig. 3** λ-orthogonality principle. Illustration of the required wavelength-dependent excitation of 4-*n*-butyl-TAD **1** (red), photopolymer **2** (green), and *o*-methylbenzophenone **3** (blue) with visible laser light (dotted) and readily available UV-emitting fluorescent lamps (dashed), respectively, to enable a λ-orthogonal system of **1** + **3** (black). All absorption spectra were recorded in chloroform at 25 °C

(refer to Supplementary Figs. 2 and 3 for visual and UV/vis analysis, respectively). Consecutive irradiation again resulted in a colorless solution without affecting the subsequent regeneration in the dark, thereby evidencing a reversibly switchable photochemical system. Similar results were obtained at lower wavelengths (down to 440 nm)—whilst keeping the number of incident photons constant—but required prolonged irradiation times (up to 30 min). The presence of polymeric **2** is supported by the characteristic peak broadening of the *n*-butyl signals in the ¹H-NMR spectrum and confirmed via high-resolution ESI-MS analysis (refer to Supplementary Figs. 4–6; Supplementary Tables 1 and 2). Furthermore, the thermal instability of **2** at ambient temperature was evidenced by the rapid decay of the polymeric signals in the resulting mass spectrum (refer to Supplementary Fig. 5). Interestingly, the photopolymerization of **1** is known to be strongly solvent-dependent[45]. Irradiation experiments in acetonitrile, for instance, do not result in the formation of **2**, as judged by the incomplete disappearance and lacking regeneration of the characteristic TAD absorption spectrum (refer to Supplementary Fig. 7). Consequently, acetonitrile solutions can be used to obtain qualitative insights concerning the photostability of **1** throughout the UV and visible light range of the spectrum (i.e., 320–560 nm), without interference of the photopolymerization reaction (refer to Supplementary Fig. 8).

In a critical next step, we probed the ability of the photochemical TAD switch from **1** to **2** to function as reaction manifold. Initially, a trapping experiment in the presence of cyclopentadiene and *N*-ethylmaleimide was performed. As expected, addition of a slight excess of cyclopentadiene (1.2 eq.) to a reference solution of **1** (0.3 M in CDCl₃) kept in the dark instantaneously resulted in the disappearance of the purple colored **1** to afford the corresponding Diels–Alder adduct **Cₚ-1**

(refer to Fig. 2a). Any non-reacted cyclopentadiene was eventually compensated for by the sequential addition of *N*-ethylmaleimide (1.4 eq.) to form a second Diels–Alder product **Cₚ-Mal** (Fig. 2a). ¹H-NMR analysis confirmed the complete consumption of **1** and **Cₚ** and the concomitant formation of the expected Diels–Alder products (Fig. 2c, spectrum I). Even after several hours, the reaction mixture did not regain any color, which supports the absence of remaining TAD.

In a sister experiment, **1** was photopolymerized at 544 nm for 10 min to quantitatively form **2**, after which 1.2 eq. of the cyclopentadiene trap was added under continuous visible light irradiation. Irradiation of the resulting mixture was then continued for an additional minute to allow for any active TAD to be trapped in the corresponding **Cₚ-1** adduct. The residual (non-reacted) diene-trap was finally quenched with 1.4 eq. of *N*-ethylmaleimide before the visible light is switched off in order to yield a non-reactive mixture that allows for offline NMR analysis (Fig. 2b). Importantly—and in contrast to the experiment without irradiation—the initially obtained colorless reaction mixture immediately regained a faint purple color upon standing in the dark, indicating the in situ regeneration of **1**. Furthermore, no trace of the **Cₚ-1** Diels–Alder adduct was observed in the ¹H-NMR spectrum (dashed red line in spectrum II, Fig. 2c), thereby unambiguously demonstrating the absence of any active TAD species upon visible light irradiation and thus the quantitative photodeactivation of **1**.

Since the above trapping experiment swiftly enables to determine the amount of reactive vs. deactivated TAD (i.e., **1** vs. **2**), the photopolymerization kinetics of **1** could now be monitored in a quantitative manner, indicating the complete deactivation of **1** to occur within 5 min (refer to Supplementary Fig. 9). Whereas photodeactivation of **1** is a relatively fast process showing pseudo-first order kinetics ($t_{1/2} = 1.7$ min in

**Fig. 4** UV-induced photoenol reaction of *o*-methylbenzophenone. Photoisomerization of *o*-methylbenzophenone **3** generates diene **4** that is susceptible toward **[4+2]**-cycloaddition reactions with electron poor enes such as *N*-maleimides. Reaction of **4** with 4-*n*-butyl-TAD **1** leads to a mixture of **5a** and **5b**, with both products shown to stand in equilibrium with one another via the labile hemiaminal type bond (highlighted in red), as was evidenced by the formation of cyclic adduct **6** upon reduction

CDCl$_3$ at 18 °C, refer to Supplementary Fig. 9), the dark time release of **1** is relatively slow with close to 4% of regenerated monomer after 2 h (at 18 °C, refer to Supplementary Fig. 10). Whereas the presence of cyclopentadiene as an in situ trap only slightly accelerated the depolymerization process (5% release of **1** after 2 h), elevated temperatures increase the depolymerization rate significantly (refer to Supplementary Fig. 11). Yet, it is crucial to establish the envisioned light-controlled reaction manifold under ambient temperature conditions in view of its applicability to surface and photoresist design.

**Orthogonality with two colors of light**. Our reaction manifold (refer to Fig. 1) entails a photochemical switch between a light-activated Diels–Alder process and a thermally induced TAD-based reaction. Thus, after having established the photochemical deactivation of TADs in the visible light regime, the λ-orthogonality between the visible light deactivation and the photochemical Diels–Alder reaction switch needs to be examined.

In order to implement the required orthogonality with two colors of light (refer to Fig. 3), we exploited the UV-induced photoenolization of *o*-methylbenzophenone (**3**, Fig. 4) to the *o*-quinodimethane isomer **4**, which swiftly reacts with electron-deficient enes (e.g., *N*-maleimides) to afford the corresponding [4+2]-cycloadduct[46, 47]. To the best of our knowledge, TADs remained unexplored dienophiles in the photoenol click reaction. However, irradiation of **3** in the presence of **1** at λ$_{max}$ = 320 nm with compact fluorescent lamps—well-studied[48] for the generation of **4** (refer to Fig. 3 and SI for emission spectra)—only resulted in minor traces of the expected [4+2]-cycloadduct **5a** (refer to Fig. 4), with the major reaction product assigned as the Alder-ene-type adduct **5b** (combined isolated yield for **5a**+**5b** of 74%, refer to SI for structure elucidation). Nonetheless, the solvent-dependent product ratio (**5a**:**5b** = 1:14 in CDCl$_3$ vs. 1:3 in DMSO-*d$_6$*) and their unsuccessful separation via column chromatography pointed toward an equilibrium between both reaction products, which is most likely established via the labile hemiaminal type bond in **5a** (highlighted in red, Fig. 4). To evidence this equilibrium, the crude reaction mixture was subjected to a triethylsilane reduction, which led to the complete conversion of both **5a** and **5b** into the exclusive product **6** (Fig. 4). As the reduction of the favored open adduct **5b** cannot directly lead to the observed cyclic **6**, the complete consumption of **5b** can only be explained via an equilibration with **5a** prior to substitution of the hydroxyl group by the mild hydride donor. Thus, although the UV-induced TAD photoenol reaction results in a mixture of two products, their dynamic interrelationship does not influence the envisaged reaction manifold since both products are exclusively generated via the same light-induced Diels–Alder reaction mode.

Having established the photostability of **1** upon UV (4 h at λ$_{max}$ = 320 nm) and of **3** upon UV (4 h at λ$_{max}$ = 320 nm) and visible light impact (1 h at 544 nm) (refer to Supplementary Figs. 12 and 13 for $^1$H-NMR spectra before and after irradiation), we combined the UV light-induced TAD photoenol reaction with the visible light-induced deactivation of **1**. Thus, an equimolar solution of **1** and **3** (0.15 M, CDCl$_3$) was first irradiated with 544 nm laser light to photodeactivate **1** via the formation of polymeric **2**. The resulting colorless reaction mixture containing **2** and **3** was next transferred into a photoreactor and subjected to UV light (λ$_{max}$ = 320 nm) to initiate the formation of the photoenol **4**. This photocaged diene immediately reacted with in situ regenerated **1** to afford the photoenol products **5a** and **5b** with 6.8% conversion after 2 h, which is in agreement with the regeneration kinetics of **1** under the exact same conditions (cf. Supplementary Fig. 10). Further, traces of a 1:2 adduct of **1** and **3** were also detected (refer to SI). When the UV light was switched off, the formation of the TAD-reactive photodiene was discontinued and a purple color unfolded in the reaction mixture due to the continued release of **1**. Importantly, close to identical results were obtained upon reversing the order of the two colors of light, thus indicating the pathway independent λ-orthogonal nature of the reaction system. However, slightly higher conversions (8.2% **5a**+**5b** after 2 h) were achieved when the photoenol reaction is carried out prior to the TAD deactivation since the rate determining step is no longer the in situ regeneration of **1**, but the photoinduced Diels–Alder reaction (see Supplementary Fig. 14 for comparative reaction kinetics).

Although we demonstrated the selective photochemical transformation of **1** in the presence of **3** and—vice versa—of **3**

**Fig. 5** Light-switchable TAD reaction manifold. After the visible light-induced deactivation of **1**, aliquots of *o*-methylbenzophenone **3** and *trans*-5-decene **7** (1:1 eq.) were added to give a non-reactive mixture (**2** + **3** + **7**, 0.15 M, CDCl₃) whilst the visible light is kept switched on. Subsequently, UV light acts as a gate to selectively switch the reaction outcome between photoenol products **5a+5b** and the thermal TAD addition product **8**

in the presence of **1** (or **2**), the formation of a deep purple [1+3] charge transfer complex was found to greatly increase the irradiation times (45 min) needed to affect the photopolymerization of **1** in the presence of **3**. It is thus favored to carry out the photodeactivation at 544 nm prior to the addition of the photoenol precursor to give a non-reactive mixture of **2** and **3**.

**Light-switchable TAD reactivity**. In a final experiment, we extended the above λ-orthogonal system by introducing a thermal TAD-based reaction mode to pioneer the light-switchable reaction manifold. *trans*-5-Decene (**7**, Fig. 5) was identified as a highly suitable thermal TAD substrate for the reactivity switch as it is stable toward both UV and visible light irradiation (refer to Supplementary Fig. 15) and gives the well-defined 1:1 Alder-ene type adduct **8** upon reaction with **1** (verified via ¹H-NMR, Supplementary Fig. 16). Furthermore, favorable reaction kinetics are associated with the formation of **8**, which are neither too fast to compete with the photoinduced Diels–Alder reaction and nor too slow to ensure for the complete trapping of any in situ regenerated TAD.

The manifold system was established by 5 min irradiation of a CDCl₃ solution of **1** at 544 nm, followed by the addition of an equimolar mixture of *o*-methylbenzophenone **3** and *trans*-5-decene **7** to provide a non-reactive one-pot reaction mixture as long as the visible light is kept switched on (refer to Fig. 5). The mixture is then subjected to UV irradiation at $\lambda_{max}$ = 320 nm for 2 h and left standing in the dark for an additional 8 h. The formation of the respective reaction products (reproduced in three-fold) was determined via integration of well resolved ¹H-NMR signals (refer to Supplementary Fig. 17). The results, depicted in Fig. 6, show the exclusive formation of photoproducts upon UV irradiation, whilst the thermal reaction is completely suppressed. Only when the photochemical reaction pathway is

blocked by placing the mixture in the dark, the formation of the thermal product **8** is initiated. Moreover, the ability to switch off both modes of reactivity is demonstrated by a 1-h period of visible light irradiation during which no TAD conversion is detected. Finally, a second on/off cycle of UV irradiation was shown to re-initiate the established reaction manifold (refer to Fig. 6 and Supplementary Fig. 32).

In summary, we demonstrate that TADs not only function as a powerful platform for small molecule and macromolecular ligation[8], yet that their thermal reactivity can be readily modulated—and indeed entirely switched off—by visible light. We exploit this photonic field modulated reactivity to switch between photochemically and thermally induced TAD reaction channels. Critically, our approach for photochemically deactivating the highly reactive TAD moiety is not only essential for modulating the outcome of TAD-based reactions by the presence or absence of light, but also hails an array of synthetic opportunities ranging from multiarea-selective surface design— where the reactivity of selected areas can be switched-off or rendered reactive by different colors of light—to advanced TAD-based photoresists for 3D laser lithography, where the on/off behavior of the TAD moiety effectively represents a depletion system potentially enabling sub-diffraction lithography[32, 33]. Thus, we submit that our pioneered light-controlled reaction manifold constitutes a step change in how light can serve as a modulator for chemical reactivity.

## Methods

**Instrumentation**. Irradiation experiments with ARIMED B6 (3 × 36 W) compact fluorescent lamps ($\lambda_{max}$ = 320 nm) were carried out in a custom-built photoreactor (refer to Supplementary Methods, Supplementary Fig. 21). Wavelength-tunable UV and visible laser light was generated with an Innolas Splitlight 600 OPO Nd: YAG Tunable Laser System using an optical parametric oscillator (OPO). The OPO is operated by a diode pumped Nd:YAG laser with a 100 Hz repetition rate. The

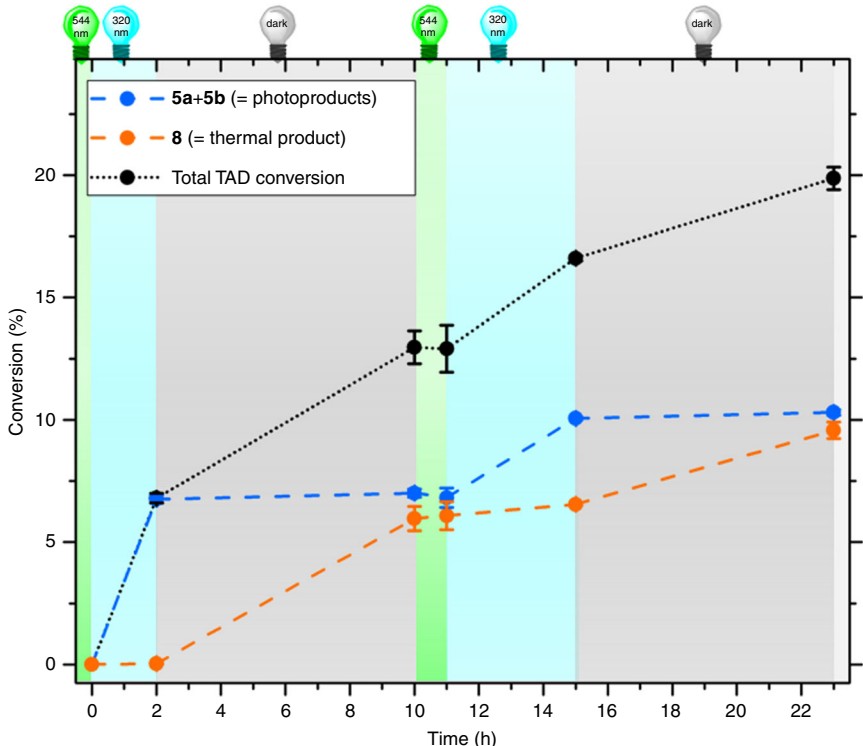

**Fig. 6** Switchable outcome of the reaction manifold. Switchable selectivity of the light-driven reaction manifold upon irradiation with UV light (2 h), followed by standing in the dark (8 h) to give photoenol products **5a+5b** and thermal addition product **8**, respectively. No TAD conversion is observed during a 1-h period of visible light irradiation, demonstrating the potential to switch off both the photochemical and the thermal TAD reaction channel. A second on/off cycle shows the possibility to re-initiate the system. The yield of the photoproducts **5a+5b**, yield of thermal adduct **8**, and the overall TAD conversion were determined via [1]H-NMR (error bars: standard deviation)

output energy was regulated by a variable attenuator (polarizer) coupled to an Energy Max PC power meter (Coherent) to measure the energy of the incident laser pulses (cf. Supplementary Table 3). The number of pulses was chosen freely and can directly be correlated to the total irradiation time. The generated laser light was guided through a prism to irradiate the bottom of the sample, which was placed in a custom-made sample holder. The temperature in the laser room was regulated at 18 °C. Emission spectra of the irradiation sources were recorded with a UV/vis SR600 spectrometer containing a polychromator and a silicon photodiode array and are depicted in Supplementary Figs. 22 and 23. UV/vis spectra were recorded on a Varian Cary 300 Bio spectrometer at 25 °C and on a Shimadzu UV-2700 spectrophotometer coupled to a CPS-100 cell positioner for thermoelectrically temperature controlled measurements. Nuclear magnetic resonance (NMR) spectra were recorded on a Bruker Avance 300 (300 MHz), Bruker Ascend 400 (400 MHz), or Bruker Avance II (500 MHz) FT-NMR spectrometer at room temperature in the solvent as indicated. Detailed spectral data for new compounds are listed below. More detailed structural elucidation is provided in the Supplementary Discussion section (i.e., Supplementary Figs. 18–20).

**Materials**. 4-*n*-butyl-TADs (**1**) was synthesized according to a literature procedure[9] followed by sublimation at 40 °C under reduced pressure ($10^{-1}$ mbar). The resulting purple crystalline product was stored in the dark at −18 °C and used within 2 weeks. All solvents and products were used as received from their supplier (cf. Supplementary Methods).

**Experimental procedures**. Detailed experimental procedures are described in the Supplementary Methods and are accompanied with reaction schemes when appropriate (Supplementary Figs. 24–31).

**Photopolymerization of 1**. A 0.3 M stock solution of **1** (69.8 mg, 0.60 mmol) in carbon tetrachloride (1.5 mL) was sealed with a septum and deoxygenated by flushing with $N_{2(g)}$ for 15 min. The stock solution was next divided into portions of 0.2 mL in crimp neck vials (0.7 mL), which were quickly flushed with $N_{2(g)}$ and crimped air-tight. The bright purple solution was then placed in the sample holder of the wavelength-tunable laser and irradiated at 544 nm (4.0 mW cm$^{-2}$, 100 Hz) to give a clear colorless solution of photopolymer **2** after 10 min. [1]H-NMR (400 MHz, CDCl₃): recorded 8 min after irradiation, $\delta$ (ppm) = 0.95 (broad m, 3H, C$H_3$), 1.40 (broad m, 3H, CH$_3$-C$H_2$), 1.68 (broad m, 3H, N-CH₂-C$H_2$), 3.31 + 3.60 (t + broad m, 2H, N-C$H_2$).

**Trapping experiment to proof the photodeactivation of 1**. An air-tight crimped headspace vial with septum containing **1** (93.1 mg, 0.60 mmol) was placed under $N_{2(g)}$ atmosphere. To this, 2.0 mL of deoxygenated (by flushing with $N_{2(g)}$ for 20 min) deuterated chloroform was added. About 0.2 mL of the resulting 0.3 M stock solution of **1** (9.31 mg, 0.060 mmol, 1 eq.) was next transferred into a crimp neck vial (0.7 mL) and quickly flushed with $N_{2(g)}$ before crimped air-tight. The sample was placed in the wavelength-tunable laser sample holder and irradiated at 544 nm (4.0 mW cm$^{-2}$, 100 Hz) for 10 min to give a clear colorless solution of **2**. To this, under continuous irradiation with visible laser light, 0.2 mL of a 0.3 M solution of cyclopentadiene (**C_p**, 4.76 mg, 0.072 mmol, 1.2 eq.) was added through the septum of the crimp neck vial. Whilst the visible light is still kept switched on, the mixture was allowed to stand for 1 min to ensure for any active TAD to react with **C_p**. Next, *N*-ethylmaleimide (10.5 mg, 0.084 mmol, 1.4 eq.) in 0.2 mL deuterated chloroform was added and allowed to react for 1 additional minute to quench any non-reacted **C_p** in the **C_p-Mal** Diels–Alder adduct before the visible light is finally switched off. The resulting mixture quickly regained a faint purple color due to the regeneration of **1**. Offline [1]H-NMR analysis (cf. Fig. 2) indicated no formation of the **C_p-1** Diels–Alder adduct and thus no active TAD species are present upon visible light irradiation, thereby demonstrating the quantitative photodeactivation of **1**.

**Photoenol reaction of 1 with 3**. A mixture of 4-*n*-butyl-TAD **1** (93.1 mg, 0.6 mmol, 1.0 eq.) and *o*-methylbenzophenone **3** (118 mg, 0.6 mmol, 1.0 eq.) in 20 mL of deoxygenated chloroform (by flushing with $N_{2(g)}$ for 10 min) was added to an air-tight capped headspace crimped vial placed under nitrogen atmosphere. The mixture was placed in a custom-built photoreactor and irradiated with 3 × 36 W ARIMED B6 compact fluorescent lamps ($\lambda_{max}$ = 320 nm) to give a clear faint yellow solution after 2.5 h. Solvent removal in vacuo yielded a yellow oil containing an equilibrium mixture of **5a** and **5b**. Purification via column chromatography (silica) failed to result in separation of the product mixture (156 mg—74%). **5a:5b** = 1:14 in CDCl₃; **5a:5b** = 1:3 in DMSO-*d₆*. [1]H-NMR (500 MHz, CDCl₃): **5b**, $\delta$ (ppm) = 0.93 (t, 3H, C$H_3$), 1.32 (m, 2H, CH$_3$-C$H_2$), 1.62 (m, 2H, N-CH₂-C$H_2$), 3.52 (t, 3H, N-C$H_2$), 4.77 (s, 2H, Ar-C$H_2$), 7.40–7.47 (m, 2H, Ar$H$), 7.48–7.54 (t, 2H, Ar$H$), 7.55–7.60 (m, 1H, Ar$H$), 7.62–7.69 (m, 2H, Ar$H$), 7.82 (m, 2H, Ar$H$), 8.43 (s, 1H, N$H$); **5a**, some resolved resonances: 4.80 (d, 1H, C$H_2$), 4.96 (d, 1H, C$H_2$). [13]C-NMR (500 MHz, CDCl₃): $\delta$ (ppm) = 13.58 (CH₃), 19.81 (CH₂), 30.02 (CH₂), 39.12 (CH₂), 47.19 (CH₂), 127.91 (CH), 128.61 (CH), 130.30 (CH), 130.74 (CH), 131.86 (CH), 132.02 (CH), 133.92 (CH), 135.15 (C), 137.04 (C), 137.62 (C), 152.99 (C), 153.38 (C), 198.55 (C). [1]H-NMR (500 MHz, DMSO-*d₆*): **5b**, $\delta$ (ppm) = 0.82 (t, 3H, C$H_3$), 1.12 (m, 2H, CH$_3$-C$H_2$), 1.39 (m, 2H, N-CH₂-C$H_2$), 3.25 (t, 3H, N-C$H_2$),

4.70 (s, 2H, Ar-C$H_2$), 7.32–7.39 (m, 1H, Ar$H$), 7.41–7.48 (m, 2H, Ar$H$), 7.50–7.61 (m, 3H, Ar$H$), 7.63–7.75 (m, 3H, Ar$H$), 10.28 (s, 1H, N$H$); **5a**, some resolved resonances: 4.74 (d, 1H, Ar-C$H_2$), 4.96 (d, 1H, Ar-C$H_2$), 6.89 (d, 1H, O$H$), 7.19 (t, 1H, Ar$H$). LC-MS (*m/z*): 352.15 [MH]$^+$. HRMS (*m/z*): $C_{20}H_{21}N_3O_3$, calc.: 352.1656, found: 352.1701 [MH]$^+$.

**Reduction of photoenol products 5a+5b into cyclic 6**. A mixture of 4-*n*-butyl-TAD **1** (93.1 mg, 0.6 mmol, 1.0 eq.) and *o*-methylbenzophenone **3** (118 mg, 0.6 mmol, 1.0 eq.) in 20 mL of deoxygenated chloroform (by flushing with $N_{2\ (g)}$ for 10 min) was added to an air-tight capped headspace crimped vial placed under nitrogen atmosphere. The mixture was placed in a custom-built photoreactor and irradiated with $3 \times 36$ W ARIMED B6 compact fluorescent lamps ($\lambda_{max} = 320$ nm) to give a clear faint yellow solution after 2.5 h. Solvent removal in vacuo afforded a yellow oil containing the equilibrium mixture of **5a** and **5b**, together with traces of unreacted **3**. The obtained crude reaction mixture was next dissolved in 5 mL of anhydrous dicholoromethane and added dropwise at 0 °C to a cooled solution of triethylsilane (1.15 mL, 7.2 mmol, 6.0 eq.) and trifluoroacetic acid in 5 mL of anhydrous dichloromethane. The mixture was allowed to react for 30 min at 0 °C, followed by overnight stirring at room temperature before being quenched with 1 M aqueous sodium hydroxide (~20 mL) to pH = 7. The yellow suspension was phase-separated and the aqueous phase washed with dichloromethane (15 mL). The combined organic phases were washed with brine (30 mL), dried over magnesium sulfate, and concentrated in vacuo. The resulting yellow oil was purified via column chromatography (silica, hexane:ethyl acetate 9:1 with a gradient to 4:1) to give unreacted **3** (14 mg—12%, $R_F$ (hexane:ethyl acetate 4:1) = 0.52) together with the cyclic reduction product **6** (169 mg—84%, $R_F$ (hexane:ethyl acetate 4:1) = 0.15) as a white waxy solid. $^1H$-NMR (500 MHz, CDCl$_3$): $\delta$ (ppm) = 0.81 (t, 3H, C$H_3$), 1.09 (m, 2H, C$H_2$-CH$_3$), 1.50 (m, 2H, N-CH$_2$-C$H_2$), 3.46 (m, 2H, N-C$H_2$-CH$_2$), 4.65 (d, 1H, Ar-C$H_2$), 5.06 (d, 1H, Ar-C$H_2$), 6.12 (s, 1H, Ar-C$H$), 7.12 (d, 1H, Ar$H$), 7.22–7.38 (m, 8H, Ar$H$). $^{13}C$-NMR (500 MHz, CDCl$_3$): $\delta$ (ppm) = 13.46 (CH$_3$), 19.41 (CH$_2$), 29.69 (CH$_2$), 38.98 (CH$_2$), 45.98 (CH$_2$), 60.03 (CH), 126.56 (CH), 127.81 (CH), 217.85 (CH), 128.45 (CH), 128.54 (CH), 128.57 (CH), 128.65 (C), 128.75 (CH), 132.75 (C), 138.32 (C), 153.62 (C), 154.62 (C). LC-MS (*m/z*): 336.10 [MH]$^+$. HRMS (*m/z*): $C_{20}H_{21}N_3O_2$, calc.: 336.1707, found: 336.1708 [MH]$^+$.

**Thermal Alder-ene addition of 1 with 7**. A solution of 4-*n*-butyl-TAD **1** (9.31 mg, 0.06 mmol, 1.0 eq.) in 0.2 mL deuterated chloroform was added to *trans*-5-decene **7** (8.42 mg, 0.06 mmol) in 0.2 mL deuterated chloroform and stirred in the dark at room temperature. Complete discoloration was observed after 20 min to give a colorless solution containing the addition product **8** in quantitative yield. $^1H$-NMR (400 MHz, CDCl$_3$): $\delta$ (ppm) = 0.84–0.97 (m, 9H, 3x C$H_3$), 1.20–1.46 (m, 8H, 3x CH$_3$-C$H_2$ + N-CH-CH$_2$-C$H_2$), 1.58–1.77 (m, 4H, N-CH-C$H_2$ + N-CH$_2$-C$H_2$), 2.01 (q, 2H, CH = CH-C$H_2$), 3.53 (t, 2H, N-C$H_2$), 4.50 (q, 1H, N-C$H$), 5.41 (m, 1H, C$H$ = CH-CH), 5.73 (m, 1H, CH = C$H$-CH), 8.53 (s, 1H, N$H$).

**Data availability**. All relevant data are available within the paper and its Supplementary Information files. All other data are available from the authors upon reasonable request.

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

## Acknowledgements

H.A.H. thanks the Research Foundation—Flanders (FWO) for the funding of his PhD fellowship. C.B.-K. acknowledges funding by the Australian Research Council (ARC) in the context of an ARC Laureate Fellowship as well as the Queensland University of Technology (QUT) for key support. Further long-term support by the Karlsruhe Institute of Technology (KIT) in the context of the Helmholtz association STN program is acknowledged. F.E.D.P. thanks Ghent University for financial support through the Concerted Research Action scheme. Kevin De Bruycker (Ghent University) and Pavleta Tzvetkova (KIT) are acknowledged for discussions regarding NMR structural elucidation. We are grateful to the QUT library as well as the QUT IFE platform "Manufacturing with Advanced Materials" for enabling the open access publication.

## Author contributions

All authors contributed to discussion and evaluation of the results at all stages. H.A.H. and C.B.-K. conceived and designed the experiments. H.A.H. performed the experiments and prepared all figures. H.A.H. wrote the manuscript in close discussion with F.E.D.P. and C.B.-K.

## Additional information

**Competing interests:** The authors declare no competing financial interests.

