## [Peer Review File · Nature Communications]

Reviewers' comments:

Reviewer #1 (Remarks to the Author):

In their manuscript „Controlling thermal reactivity with different colors of light“ Houck et.al report on a clever one-pot system that shows different reactivities depending on the input of different light signals.

More precisely, in this work the reactivity of TADs is effectively steered into different reaction channels via irradiation with light of different wavelengths in the presence of o-methyl-benzophenone. Visible light irradiation leads to polymerization of TADs and irradiation with UV-light gives Diels-Alder-type products with the benzophenone. This establishes a lambda-orthogonal reaction scheme, which is then expanded with a third reaction channel, i.e. the thermal addition reaction of TADs with trans-5-decene. This latter thermal reaction can be halted and started again using light as outside control element.

Overall this work presents a carefully designed system that significantly expands the levels of outside control over chemical reactivity and should therefore be of great interest for the broad readership of Nature Communications. The concept is convincing and very innovative, the presented system explores new chemistry of TADs and is well chosen to demonstrate effective outside control over the thermal reactivity within a complex regulated reaction setting.

I therefore recommend publication but I have a couple of points to be addressed before:

1. One issue I have concerns the low conversions reported for the UV-light triggered Diels-Alder reaction between TAD 1 and the benzophenone 3. Quite high concentrations are used (0.3 M) in the experiments, which limits light penetration and “uncaging” of the diene 4. After 2 h of irradiation only about 7% yield is obtained in total. What is the maximum yield that can be obtained from this reaction? In the supporting information the reported yield is 74%? It can also be seen in the repetition experiment, that this yield drops significantly in the second run (Figure 6). How many more runs can be done before the system deteriorates? Does a reduction in the total concentration enhance the performance?
2. When discussing the orthogonality of the light-reactions of 1 and 3, the photochemistry of the TAD polymer 2 should be included as well. This is only mentioned shortly before (Figure 1) but could have a strong influence on the reaction manifold. What is the extinction/relative absorbance of the polymer 2 with respect to 1 and 3 (Figure 3)?
3. Along similar lines, in Figure 3 the extinction coefficients or at least the relative absorbances of the different absorbing species in the mix should be given instead of normalized spectra. For the moment it seems strange that the diene-deprotection at 320 nm can proceed at all, given the seemingly much higher absorption of TAD in the UV part of the spectrum.
4. How fast is the UV-light induced depolymerization of 2 under the irradiation conditions? The experiment is mentioned in the supporting information but without details.
5. What is the maximum yield of the addition reaction between TAD 1 and trans-5-decene under thermal conditions?

Minor points:

- Line 31: it is a little hard to understand the sentence. Do the authors mean that light induced methods provide greater temporal resolution and are the only methods to provide spatial control as opposed to thermal triggers?

- Line 53, there are some precedents for light control of thermal reactivity see for instance Kathan et. Al, Angew. Chem. Int. Ed. 2016, 55, 13882 or Samachetty et al. Tetrahedron, 2008, 64, 8292.

- line 57 – “thereby providing a selective on- and off- switch over their thermal reactivity” – what is the selective on-switch for the thermal reactivity? The slow thermal depolymerization in the dark or the UV-light triggered depolymerization?

-starting line 92: I am not sure how much information about photostability in the reaction medium CDCl₃ can be gained from the experiment about the photostability in MeCN. I would therefore move this experiment to the Supporting Information. What is meant by “non-fluorescing solvent”?

- line 145 – the authors should give the combined yield of 5a and 5b already here.

Reviewer #2 (Remarks to the Author):

This is an interesting chemical/photochemical multistructural system that allows the authors to control several aspects of triazolinedione (TAD) reactivity. The work uses a selection of known TAD thermal and photochemical reactions - the novelty is their assembling these into a multicomponent system. It is not completely clear that these results will be generalizable to practical applications (as the authors postulate). In a communication it is challenging to cover all the pertinent details of control experiments, etc., but there are multiple issues that should be addressed before publication, as follow:

(1) Some of the photochemistry is done at approximately 320 nm (really 300 - 400 nm according to Figure 3). It appears that the TAD 1 absorbs strongly in this region also. How are the concentrations adjusted so that 3 mainly absorbs the light, vs. TAD 1. How much photopolymerization does 1 do under these conditions?

(2) How reversible is the TAD photopolymerization over many cycles? The authors seem to indicate incomplete recovery of TAD from the polymer, which doesn't seem compatible with the propositions of "...well defined off-switch..." and "...modulation of TAD's reactivity by light is completely reversible...."

(3) In figure 1 there is a reverse arrow from 2 back to 1 that seems to indicate that light can also reverse the polymerization - is that observed? What wavelength?

(4) The discussion in lines 94 to 98 is unclear. I do not understand what the authors mean by "...assess the photostability of 1 from 320 to 560nm....". Perhaps they could clarify the relevance of this control experiment. Does anyone know why acetonitrile quenches the photopolymerization of 1? The reference 42 by Pirkle just describes aromatic solvents as excited state quenchers of alkyl TADS.

(5) The statement on line 160 "...established the photostability of 1 and 3..." is incorrect. Do they mean just the photostability of 3 (as shown in figure S11)?

(6) The thermal reversion of the TAD sequestering polymer 2 seems to complicate control in this system. For example, they note that the warmer temperatures in the photoreactor appeared to cause more TAD back formation. It seems that there should be a little bit of exploration of how different temperatures affect these results, or how temperature is another variable that can/should be controlled.

(7) The citation in ref. 33 appears incomplete.

Reviewer #3 (Remarks to the Author):

This work reports a very fundamental study of TAD chemistry driven by light at different

wavelength. The light induced deactivation facilitates the TAD chemistry on demand. Moreover, it also provides possibility to realize this process as a photochemical ligation which can be combined with the other process. I believe this work will find wide interest from both polymer and material science. Particular advantage comes from the fact that deactivation and reactivation can be triggered by red light and blue light and thermal processes. The proposed ideas are proven by experimental evidences. All the intermediates and final products are well characterized by spectroscopic analysis. It can be accepted as is after considering the following minor comment.

The deactivation occurs by red light irradiation through the formation of polymer. The formed polymer has an absorption at lower wavelengths. The authors indicate that it can be activated thermally or by blue light. Photoenolization reaction also occurs at the blue light region. Thus, the absorption of two components may interfere and reduce the activity of the independent reaction. Apparently, no problem exists when the activation is made thermally. This point should be clarified.

The title does not reflect the actual process. It only says control of general thermal activity. There should also be some information on the actual chemistry involved.

Response to referees' comments – NCOMMS-17-18140

Referee #1 (Remarks to the Author):

1. One issue I have concerns the low conversions reported for the UV-light triggered Diels-Alder reaction between TAD 1 and the benzophenone 3. Quite high concentrations are used (0.3 M) in the experiments, which limits light penetration and “uncaging” of the diene 4. After 2 h of irradiation only about 7% yield is obtained in total. What is the maximum yield that can be obtained from this reaction? In the supporting information the reported yield is 74%? It can also be seen in the repetition experiment, that this yield drops significantly in the second run (Figure 6). How many more runs can be done before the system deteriorates? Does a reduction in the total concentration enhance the performance?

The concentrations used are indeed high for photochemical reactions. However, the obtained maximum yield for the reaction of **1** with **3** upon UV-irradiation is 74% (isolated yield) and is in line with the yields reported for the photo-induced Diels-Alder reaction of **3** with *N*-maleimides (i.e. 75-80%, doi: 10.1002/ange.201509472).

For the reaction manifold itself, the slow depolymerization and *in situ* regeneration of TAD directly affects the obtained conversions. Since the reaction mixture does not show a pink color, no free monomer is present throughout the experiments. In other words, the liberated TAD immediately reacts with the photochemical or thermal reaction partner. Figure S10 indicates that after 2h under UV-light conditions, approximately 7% of TAD is regenerated, which exactly matches the 7% conversion of the photoproducts within 2h as seen in Figure 6. The overall decrease in conversion over time can thus be attributed to the slower depolymerization, which is a thermal process and therefore slows down when the effective concentration of the polymer decreases over time.

Although a reduction in the total concentration of the reaction manifold will undoubtedly have an accelerating effect on the photochemically induced reaction channel (more light penetration), the depolymerization – which is the rate determining step throughout the manifold – is a thermal event and will proceed slower in more diluted conditions. As a result, lower overall conversions of both the photoproducts and the thermal reaction products will be observed.

2. When discussing the orthogonality of the light-reactions of 1 and 3, the photochemistry of the TAD polymer 2 should be included as well. This is only mentioned shortly before (Figure 1) but could have a strong influence on the reaction manifold. What is the extinction/relative absorbance of the polymer 2 with respect to 1 and 3 (Figure 3)?

The absorption spectrum of the TAD photopolymer **2** has now been included in Figure 3 (see below). It can be seen that the absorption in the visible range is completely absent due to the disappearance of the N=N double bond of **1** as a result of its incorporation into the polymer backbone. Furthermore, the absorption of the photopolymer in the UV-range has decreased significantly, which can be attributed to the loss of conjugation of the carbonyl groups on the N-backbone. Hence, the absorbance of the photopolymer with regard to the emission sources is very limited and therefore its influence on the reaction manifold is negligible.

Figure 3 | λ -orthogonality principle. Illustration of the required wavelength-dependent excitation of 4-*n*-butyl-TAD **1** (red), photopolymer **2** (green) and *o*-methylbenzophenone **3** (blue) with visible laser light (dotted) and readily available UV-emitting fluorescent lamps (dashed), respectively, to enable a λ -orthogonal system of **1+3** (black). All absorption spectra were recorded in chloroform at 25 °C.

In addition, we noted that the solvent in which the absorption spectra are measured was not mentioned in the caption of Figure 3. This relevant information has now been added.

3. Along similar lines, in Figure 3 the extinction coefficients or at least the relative absorbances of the different absorbing species in the mix should be given instead of normalized spectra. For the moment it seems strange that the diene-deprotection at 320 nm can proceed at all, given the seemingly much higher absorption of TAD in the UV part of the spectrum.

Indeed, the spectra of TAD and the photoenol precursor have been normalized individually and are therefore not representative for the overall reaction mixture. To correct for this, the spectra of the starting reagents have now been normalized, taking into account their relative concentrations (refer to Figure 3 in the manuscript and below). Furthermore, the absorption spectrum of an equimolar mixture of TAD and *o*-methylbenzophenone **1+3** has been measured and is now included. With the latter, it is evident that the absorption of the TAD species in the UV part of the spectrum is much lower compared to the *o*-methylbenzophenone and hence does not prevent the diene-deprotection reaction to occur at 320 nm.

4. How fast is the UV-light induced depolymerization of 2 under the irradiation conditions? The experiment is mentioned in the supporting information but without details.

The results of the investigated depolymerization/regeneration kinetics are presented in Figure S10. A cross-reference to the obtained kinetics is made within the revised manuscript and has also been included in the appropriate experimental section.

5. What is the maximum yield of the addition reaction between TAD **1 and *trans*-5-decene under thermal conditions?**

The thermal addition reaction of TAD with *trans*-5-decene is categorized as a click reaction and proceeds quantitatively. This information has now been added to the procedure in the Supporting Information section.

Minor points:

- Line 31: it is a little hard to understand the sentence. Do the authors mean that light induced methods provide greater temporal resolution and are the only methods to provide spatial control as opposed to thermal triggers?

We rephrased the sentence in order to emphasize that light is a more defined trigger to enable spatio-temporal resolution compared to its thermal triggered analogues.

- Line 53, there are some precedents for light control of thermal reactivity see for instance Kathan et. Al, Angew. Chem. Int. Ed. 2016, 55, 13882 or Samachetty et al. Tetrahedron, 2008, 64, 8292.

The proposed citations indeed represent good examples of how light can be used to regulate reactivity. Thus, these references have been included and are shortly discussed in the 3rd paragraph of the introduction. It is highlighted that although both studies demonstrate that the thermal reactivity of the central molecules can be modulated (i.e. accelerated or suppressed), the thermal reactivity of both formed isomers as such can never be completely and fundamentally halted. Our pioneered reaction manifold thus surpasses these existing studies since the visible light transformation of TAD results in the quantitative deactivation and therefore offers a true off-switch over their thermal reactivity.

- line 57 – “thereby providing a selective on- and off- switch over their thermal reactivity” – what is the selective on-switch for the thermal reactivity? The slow thermal depolymerization in the dark or the UV-light triggered depolymerization?

The on-switch for the thermal reactivity is obtained by the depolymerization, which is affected by switching off the visible light source. UV-light does not trigger the depolymerization reaction, but is essential to induce the formation of the diene reaction partner to affect the photochemical reaction pathway.

-starting line 92: I am not sure how much information about photostability in the reaction medium CDCl₃ can be gained from the experiment about the photostability in MeCN. I would therefore move this experiment to the Supporting Information. What is meant by “non-fluorescing solvent”?

We agree that this experiment carried out in a different solvent only allows for a qualitative assessment to be made concerning the photostability of **1**. Taking into account the remark made by reviewer #2 on this section, lines 92-98 have been reworded to better focus the reader’s attention on the strong solvent dependence of the photodeactivation.

- line 145 – the authors should give the combined yield of 5a and 5b already here.

The combined yield has been stated in the text.

Referee #2 (Remarks to the Author):

1. Some of the photochemistry is done at approximately 320 nm (really 300 - 400 nm according to Figure 3). It appears that the TAD 1 absorbs strongly in this region also. How are the concentrations adjusted so that 3 mainly absorbs the light, vs. TAD 1. How much photopolymerization does 1 do under these conditions?

Indeed, from Figure 3 it appears that TAD 1 strongly absorbs in the UV-region of the spectrum – a point that has also been made by reviewers #1 and #3. In the original Figure 3, the spectra of TAD and the photoenol precursor have been normalized individually and are therefore not representative for the overall reaction mixture 1+3. This has been corrected and the spectra of 1+3 has been implemented into Figure 3 (*vide supra*). From the absorption of the mixture, it is now clear that the absorption arising from the TAD species in the UV part of the spectrum is much lower with regard to *o*-methylbenzophenone and therefore does not significantly interfere with the formation of the photo-diene at 320 nm.

The concentration used to affect the photopolymerization of 1 (i.e. 0.3 M) is taken from the initial 1970 reference of Pirkle. After irradiation, an equimolar amount of a 0.3 M solution of 3 is added, giving a 0.15 M total concentration of 1+3 (doubled volume).

Figure S10 shows the depolymerization/regeneration of 1 over time, immediately after visible light irradiation. From this, a faster depolymerization is observed under the applied conditions of UV-irradiation with respect to its dark reference. No photopolymerization is observed when 1 is irradiated with UV-light.

2. How reversible is the TAD photopolymerization over many cycles? The authors seem to indicate incomplete recovery of TAD from the polymer, which doesn't seem compatible with the propositions of "...well defined off-switch..." and "...modulation of TAD's reactivity by light is completely reversible...."

The complete recovery of TAD from the photopolymer is a function of the waiting time and sufficiently long times will lead to a complete TAD recovery. However, the question of complete TAD recovery or even its rate is non-critical for the performance of the light controlled reaction manifold. The relevant reaction that defines the effectiveness ("well-defined") of the off-switch is the fast transformation of the TAD into its polymeric form. This switch is efficient and quantitative, removing all free TAD from the solution by irradiation with visible light. However, we agree that the term "completely reversible" with regard to the TAD regeneration is maybe too strong as we cannot exclude the possibility that not all TAD is recovered – however irrelevant this process is for our light controlled manifold. We thus removed the term "completely" throughout the manuscript whenever related to the reversible nature of TAD regeneration.

3. In figure 1 there is a reverse arrow from 2 back to 1 that seems to indicate that light can also reverse the polymerization - is that observed? What wavelength?

The photopolymerization reaction is only observed upon irradiation with light from the visible part of the spectrum. Under UV-conditions, there is no driving force toward the formation of the polymer and therefore the depolymerization process is not affected under these conditions. We wanted to highlight the continued depolymerization under UV-light by the addition of " λ_{UV} " to the reverse arrow.

We realize, however, that this maybe confusing and have therefore deleted " λ_{UV} " in the reaction scheme of Figure 1.

4. The discussion in lines 94 to 98 is unclear. I do not understand what the authors mean by "...assess the photostability of 1 from 320 to 560nm...". Perhaps they could clarify the relevance of this control experiment. Does anyone know why acetonitrile quenches the photopolymerization of 1? The reference 42 by Pirkle just describes aromatic solvents as excited state quenchers of alkyl TADS.

The reference of Pirkle indeed states that it is essential that the excited state of TAD is not quenched in order to allow for its photopolymerization to occur. It is believed that solvents in which TADs do not fluoresce, severely decrease the life time of their excited state and thereby inhibit their photopolymerization. Acetonitrile is a solvent in which 4-alkyl-TADs do not fluoresce and was indeed found to inhibit the formation of the photopolymer upon visible light irradiation.

Following this observation, irradiation experiments of TAD solutions in acetonitrile can offer valuable information concerning the photostability of TADs. This stability is readily assessed by the comparison of the UV/vis spectra before and after irradiation – since no photopolymerization can account for the decrease in absorption. Since our reaction manifold exploits the reactivity of TADs under both UV- and visible light irradiation conditions, it was necessary to initially obtain qualitative insight with regard to their stability in these regions of the spectrum and therefore blank irradiation experiments were carried out at well-defined intervals from 320 nm up to 560 nm.

We do, however, accept that the description of this control experiment is unclear, which has also been pointed out by reviewer #1. We have thus rephrased lines 92-98 to focus the reader's attention more on the strong solvent dependence of the photopolymerization of **1**.

5. The statement on line 160 "...established the photostability of 1 and 3..." is incorrect. Do they mean just the photostability of 3 (as shown in figure S11)?

The stability of **1** upon UV-irradiation (ARIMED B6, 3x36W, 4h) was investigated, which was analyzed via ¹H-NMR spectroscopy. To facilitate the analysis, a diene trap was added to form a Diels-Alder adduct. The NMR spectrum after irradiation is identical to a reference sample kept in the dark, thus indicating the photostability of **1**.

We greatly appreciate bringing to our attention that the respective NMR spectra of this control experiment were not included into the Supporting Information section. This was rectified by the addition of Figure S13, included here below, along with a cross-reference in the manuscript.

Supplementary Figure S13 | $^1\text{H-NMR}$ spectra ($\text{DMSO-}d_6$) of **1** after a addition of *trans,trans*-2,4-hexadien-1-ol (1.1 eq) before (a) and after irradiation with UV-light ($\lambda_{\text{max}} = 320 \text{ nm}$, ARIMED B6, 3 x 36W, 4 h), indicating no photodegradation. Residual signals can be attributed to the slight excess of the diene (c). Spectra were measured in $\text{DMSO-}d_6$ to ensure better solubility of the potentially formed degradation products.

6. The thermal reversion of the TAD sequestering polymer **2 seems to complicate control in this system. For example, they note that the warmer temperatures in the photoreactor appeared to cause more TAD back formation. It seems that there should be a little bit of exploration of how different temperatures affect these results, or how temperature is another variable that can/should be controlled.**

The depolymerization of **2** is indeed a thermal process which will be faster at higher temperatures and is caused by the typically low ceiling temperature that such all N-backbone polymers possess.

We agree that this claim requires additional data. Thus, we have carried out additional experiments in which the regeneration of the monomer **1** is monitored via UV/vis spectroscopy at different temperatures (i.e. at 18 °C, 25 °C and 35 °C). The results, added as Supplementary Figure S11 and depicted here below, indeed show a strong temperature dependence in the release of **1** with significantly faster depolymerization at higher temperatures. Since this depolymerization is the rate determining step within our reaction manifold, it is thus expected that similar conversions will be reached faster. Nonetheless, we sought it of critical importance that the designed manifold is established at ambient temperature conditions to maintain its potential in controlled surface and photoresist applications. The latter, together with a cross-reference to Figure S11, is now highlighted in the revised manuscript text.

Supplementary Figure S11 | Kinetic UV/vis spectra (CHCl_3) recorded after 5 minutes of visible light irradiation (544 nm, 4 mW cm^{-2} , 100 Hz) of a 0.3 M solution of **1** in CHCl_3 showing the regeneration of **1** at different temperatures, i.e. 18 °C (blue), 25 °C (black) and 35 °C (red).

7. The citation in ref. 33 appears incomplete.

Changed accordingly.

Referee #3 (Remarks to the Author):

1. The deactivation occurs by red light irradiation through the formation of polymer. The formed polymer has an absorption at lower wavelengths. The authors indicate that it can be activated thermally or by blue light. Photoenolization reaction also occurs at the blue light region. Thus, the absorption of two components may interfere and reduce the activity of the independent reaction. Apparently, no problem exists when the activation is made thermally. This point should be clarified.

This is indeed a good point that has also been made by reviewers #1 and #2. The problem was addressed by implementing the absorption spectrum of the reaction mixture **1+3** in Figure 3 (*vide supra*), from which it is clear that the absorption of the photoenol precursor is much higher than of the TAD and/or photopolymer. We can therefore anticipate a rather low interference with the UV-induced formation of the diene.

2. The title does not reflect the actual process. It only says control of general thermal activity. There should also be some information on the actual chemistry involved.

With the title, we aimed to stress the unique and unprecedented ability to completely switch off a molecule's thermal reactivity by applying an external photonic field. Furthermore, we hope that via this journal our study can reach a broad audience and can serve as a basis in the future development of reaction manifolds with precision outer field control. We feel that a title containing more details might discourage non-specialized readers and therefore not transmit these intentions.

REVIEWERS' COMMENTS:

Reviewer #1 (Remarks to the Author):

The authors have thoroughly revised their manuscript and have answered all questions satisfactorily. They have added all necessary missing information and addressed the critical points. I therefore recommend publication in Nature Communications.

Reviewer #2 (Remarks to the Author):

The authors have now done a good job addressing the concerns of all 3 reviewers, myself included. The re-write of the manuscript is now much more clear, especially regarding several confusing text sections.

I believe that now the paper is ready for publication. I also agree that it is appropriate for the broad audience of Nature. One value is that I believe that this work will spur other investigators to explore new multi-photochemical/thermal reaction systems.

Reviewer #3 (Remarks to the Author):

The authors' response to my comments are satisfactory. I, therefore, suggest its publication.

Response to referees' comments on our revised manuscript NCOMMS-17-18140A

We are pleased to hear all referees are satisfied with our revised manuscript and have no additional response to their comments (included below).

Referee #1 (Remarks to the Author):

The authors have thoroughly revised their manuscript and have answered all questions satisfactorily. They have added all necessary missing information and addressed the critical points. I therefore recommend publication in Nature Communications.

Referee #2 (Remarks to the Author):

The authors have now done a good job addressing the concerns of all 3 reviewers, myself included. The re-write of the manuscript is now much more clear, especially regarding several confusing text sections.

I believe that now the paper is ready for publication. I also agree that it is appropriate for the broad audience of Nature. One value is that I believe that this work will spur other investigators to explore new multi-photochemical/thermal reaction systems.

Referee #3 (Remarks to the Author):

The authors' response to my comments are satisfactory. I, therefore, suggest its publication.